# Nuclear Medicine and Molecular Imaging in Urothelial Cancer: Current Status and Future Directions

**DOI:** 10.3390/cancers17020232

**Published:** 2025-01-13

**Authors:** Sam McDonald, Kevin G. Keane, Richard Gauci, Dickon Hayne

**Affiliations:** 1Urology Department, South Metropolitan Health Service, Murdoch, WA, 6150, Australia; sam.mcdonald@health.wa.gov.au (S.M.); kevin.keane@health.wa.gov.au (K.G.K.); richard.gauci@health.wa.gov.au (R.G.); 2UWA Medical School, University of Western Australia, Crawley, WA 6009, Australia; 3Australian and New Zealand Urogenital and Prostate (ANZUP) Cancer Trials Group, Sydney, NSW 2000, Australia

**Keywords:** urothelial cancer, PET imaging, immunoPET, theranostics

## Abstract

Urothelial cancer is one of the most common urological malignancies and can occur in both the upper and lower urinary tract. It is a costly disease, and both incidence and mortality are rising around the world. Accurate imaging is crucial for diagnosis, staging, and treatment. Conventional imaging techniques have limitations in this regard, and Positron Emission Tomography (PET) imaging is becoming increasingly used in this disease. This paper explores the current role of nuclear medicine and molecular imaging in urothelial cancer, with a special emphasis on potential future advancements including emerging novel PET tracers and techniques. The authors aim to enhance current knowledge on the topic and the future directions of nuclear medicine in the field of urothelial cancer.

## 1. Introduction

Urothelial carcinoma is the second most common urological malignancy in developed countries, with rising incidence and mortality [1]. Urothelial carcinoma can occur in both the lower urinary tract (urethra/bladder) and the upper urinary tract (renal pelvis/ureter). Bladder cancer comprises 90–95% of urothelial cancers, while upper tract urothelial carcinoma (UTUC) represents 5–10% [2,3]. Bladder cancer is the 9th most commonly diagnosed cancer globally and has the highest lifetime treatment cost per patient of all cancers [2,4]. An estimated 614,298 people were diagnosed with the disease globally in 2022, with the annual number of new cases and deaths from bladder cancer predicted to rise dramatically by 2040 [4,5].

Positron Emission Tomography is an imaging technology widely used in oncology, and is a mainstay in the clinical management of many cancers, including lung, lymphoma, breast, and melanoma. However, its role in urothelial cancer has been less defined and somewhat controversial.

Traditionally, staging in urothelial cancer has relied on contrast-enhanced computed tomography (CT) of the chest, abdomen, and pelvis [6]. Bone scans have also been used historically, but are now questioned for their validity in this setting, particularly prior to cystectomy [7]. Despite the prominence of CT imaging, its limitations in detecting certain metastatic or recurrent sites have driven interest in more accurate imaging modalities, including PET.

In recent years, ^18^F-fluorodeoxyglucose (^18^F FDG) PET-CT has gained traction for staging muscle invasive bladder cancer (MIBC), especially in cases where conventional imaging results are inconclusive [6,8]. PET-CT is increasingly used to evaluate lymph node involvement and assess for recurrence after treatment.

With recent significant advancements in treating locally advanced and metastatic bladder cancer, particularly with enfortumab vedotin and pembrolizumab, precise staging in urothelial cancer has become critical [9]. Early identification of low-volume metastatic disease can guide patients toward the most suitable systemic therapies, sparing them from invasive, morbid, and potentially futile operations.

This article aims to explore the current landscape of molecular imaging in the clinical management of urothelial cancer, with a focus on primary staging, response evaluation, and recurrence detection. Special emphasis will be placed on the role of ^18^F FDG PET-CT in MIBC, as well as potential future advancements including emerging non-^18^F FDG PET agents, PET radiopharmaceuticals, and PET-MRI applications.

## 2. Materials and Methods

This review was developed adhering to the Scale for the Assessment of Narrative Review Articles (SANRA) guidelines. We examined articles covering the following topics: ^18^F FDG PET-CT in urothelial cancer, non-^18^F FDG PET agents in urothelial cancer, theranostics, and new advances in molecular imaging in urothelial malignancy. A thorough literature search was conducted via the PubMed database to identify studies relevant to article scope. Search terms included “PET”, “PET-CT”, “PET-MRI”, “Positron Emission Tomography”, “^18^F FDG PET-CT”, “Urothelial Cancer”, “Upper Tract Urothelial Cancer”, “non-^18^F FDG PET” and “Theranostics”. We identified 1292 articles. Non-English language articles, off-topic papers, and case reports were excluded, resulting in 80 articles being selected for discussion. Two authors conducted the selection process to ensure impartial decisions.

## 3. What Is PET?

PET is a nuclear medicine imaging technique that provides detailed metabolic and functional information about tissues and organs. PET works by using a radiolabelled tracer, typically a biologically active molecule labelled with a positron-emitting isotope. The tracer accumulates in target tissues and emitted positrons collide with electrons, producing gamma photons travelling in opposite directions. This unique photon generation, when detected by the PET camera and analysed, allows precise localisation of the tracer and its concentration, and can reveal metabolic activity at the molecular level [10,11]. In PET imaging, SUV max (Standardised Uptake Value maximum) is a quantitative metric that measures the highest concentration of a radiotracer (such as FDG) in a region of interest within the body. The SUV max is a way to assess how much of the tracer is taken up by the most malignant tissue and is calculated by normalising the uptake to the injected dose and patient body weight.

Different tracers may be used in PET depending on the metabolic or cellular processes that need to be evaluated, providing detailed insight into the biochemical landscape of the disease. Table 1 demonstrates the ideal characteristics of a successful PET tracer for urothelial cancer. Fluorine-18 (^18^F), Oxygen-15 (^15^O), Carbon-11 (^11^C), Nitrogen-13 (^13^N), and Gallium-68 (^68^Ga) are all commonly used positron emitting radioisotopes. PET-CT merges functional information from PET with anatomical detail from CT. This combination enhances diagnostic accuracy, allowing for precise localisation of abnormal metabolic activity.

## 4. The Role of ^18^F FDG PET-CT in Urothelial Cancer

### 4.1. Primary Lesion

The fundamental aim of primary lesion staging in urothelial cancer is to determine if the tumour (T) is non-muscle-invasive (CIS, Ta, T1), or muscle-invasive (T2 or above), which will alter the treatment pathway significantly. Traditionally, surgical interventions such as transurethral resection of the bladder tumour (TURBT) and uretero-renoscopy have been employed to obtain a histological diagnosis, grade, and stage in urothelial carcinoma. However, these procedures are intrusive and not without risk. Furthermore, accuracy of surgical staging in urothelial tumours has been shown to vary between urologists, with a third of high-risk non-muscle-invasive bladder cancers being upgraded to muscle invasive disease following re-resection [12]. Efforts have been made to develop less invasive imaging techniques for T-staging in urothelial malignancy. Multiparametric MRI to evaluate T-stage in bladder cancer is becoming increasingly widespread with the development of the vesical imaging-reporting and data system (VIRADS) [13]. However, appropriate staging for UTUC remains a dilemma due to the challenge of obtaining adequate tissue sampling with ureteroscopy. Although not currently recommended by guidelines, there is some evidence of improved prognosis for UTUC patients with T3+ or node-positive tumours treated with neoadjuvant chemotherapy (NAC), underscoring the importance of accurate T-staging at diagnosis [14].

^18^F FDG PET-CT is limited in evaluating primary lesions in urothelial cancer due to the urinary excretion of the tracer, which may obscure tumours in the renal pelvis, ureter, and bladder. Despite several proposed methods to limit tracer accumulation in the urine, including oral hydration, urinary catheterisation, bladder irrigation, delayed image sequencing, and forced diuresis, conventional imaging strategies such as CT and MRI remain superior [15].

Summary: ^18^F FDG PET-CT has no established role in evaluation of a primary urothelial tumour.

### 4.2. Regional Staging

Lymph node involvement is common in muscle-invasive bladder cancer, with 25% of T2 and 50% of T3 tumours metastasising to regional lymph nodes [16,17]. Similarly, in UTUC, lymph node involvement is seen in 20–30% of UTUC cases [18,19]. CT and MRI are commonly used to assess for lymph node metastasis in patients with urothelial carcinoma. They rely primarily on size criteria, with current recommendations suggesting a cutoff of ≥10 mm for retroperitoneal lymph nodes and ≥8 mm for pelvic lymph nodes to indicate suspicion for metastasis [20,21]. However, these size-based guidelines have limitations, as bladder cancer metastases often involve lymph nodes that show minimal or no enlargement, and there exists significant interobserver variation leading to high false-negative rates for both CT and MRI [16]. Accuracy for lymph node involvement in bladder cancer is reported to range from 73% to 92% for CT and MRI [22].

Comparatively, sensitivity for ^18^F FDG PET-CT in detection of lymph node metastases in bladder cancer ranges broadly, from 23% to 100%, and specificity from 33% to 100%, likely accounted for by variability in patient populations, heterogenous study designs, and imaging protocol and interpretation differences, underscoring the need for further standardisation to improve diagnostic accuracy in lymph node staging for bladder cancer [23].

Most studies report that ^18^F FDG PET-CT has higher sensitivity, with similar specificity for detecting metastatic lymph nodes when compared to conventional imaging [24,25,26,27]. A systematic review and meta-analysis of 35 articles by Crozier et al. concluded that ^18^F FDG PET-CT provides superior sensitivity compared to CT (56% vs. 40%), and similar specificity for detecting positive lymph nodes in bladder cancer prior to cystectomy [25].

Limited data exist regarding optimal nodal imaging in UTUC due to the low sensitivity of conventional imaging in detecting lymph node metastases in this setting, and lymph node dissection during radical nephroureterectomy for high-risk disease remains the standard of care [28,29]. Several promising studies suggest that ^18^F FDG PET-CT is beneficial in this setting [30,31]. Voskuilen et al. reported ^18^F FDG PET-CT had 82% sensitivity and 84% specificity for detecting lymph node metastasis in patients with UTUC. They also found that suspicious lymph nodes on ^18^F FDG PET-CT were associated with worse recurrence-free survival [31]. Overall ^18^F FDG PET-CT seems to have high sensitivity for lymph node staging in UTUC, with a systematic review in 2021 showing that sensitivity ranged between 82% and 95%, with a specificity of 84–91% [32].

Summary: ^18^F FDG PET-CT shows an improvement in sensitivity for detecting pelvic lymph node involvement in bladder cancer compared to CT alone. For UTUC patients, ^18^F FDG PET-CT shows promise in N-staging, although further research is needed to clarify its role in this group.

### 4.3. Distant Metastases

Distant metastases are commonly encountered in patients with invasive urothelial cancer. Accurate staging is crucial to optimise treatment strategies, as early detection of metastases can prevent unnecessary major surgery and guide appropriate therapy [33]. ^18^F FDG PET-CT is being increasingly used clinically in the evaluation of distant metastases in bladder cancer. Several reviews highlight the superiority of ^18^F FDG PET-CT over CT alone [27,34,35]. A retrospective study by Ozturk et al. on 79 patients reported a sensitivity of 89% and a specificity of 78%, with positive predictive and negative predictive values of 90% and 75%, respectively [36]. This is consistent with findings from Lu et al., who reported an 89% sensitivity and 82% specificity for detecting metastatic disease using ^18^F FDG PET-CT [37]. However, despite increasing clinical use of ^18^F FDG PET-CT to assess for distant metastases in urothelial cancer, evaluation of advantages and pitfalls of this imaging modality compared to CT alone is not yet complete [8].

Summary: ^18^F FDG PET-CT has shown superior performance in detection of distant metastases in urothelial carcinoma compared to CT/MRI in some studies, and is being increasingly used clinically in this setting. However, its use is not currently supported by guidelines.

### 4.4. Assessing Treatment Response

Cisplatin-based treatment is the recommended NAC for MIBC before radical cystectomy in eligible patients, with those achieving a complete pathological response demonstrating significantly better outcomes [6]. Conventional imaging methods like CT and MRI struggle to distinguish viable tumour cells from necrotic tissue or identify small areas of micro-metastatic disease, limiting their effectiveness in assessing NAC response. Although current guidelines do not recommend ^18^F FDG PET-CT for treatment response assessment, studies suggest it has potential in this setting [38].

Higashiyama et al. reported a sensitivity of 92% for ^18^F FDG PET-CT in detecting residual invasive bladder cancer post-NAC, while Soubra et al. found that it demonstrated a 78.5% sensitivity and 95.6% specificity in identifying complete pathologic response [39,40].

A recent study evaluated ^18^F FDG PET-CT for nodal response assessment in MIBC patients receiving neoadjuvant pembrolizumab in the PURE-01 trial. The study found that while ^18^F FDG PET-CT had limited sensitivity for detecting lymph node metastasis (27% pre-treatment and 37.5% post-treatment), it showed high specificity (97% and 98% pre- and post-treatment, respectively) and helped predict lymph node involvement. The authors concluded that baseline ^18^F FDG PET-CT could assist in selecting patients best suited for neoadjuvant immunotherapy in a trial setting [41].

Summary: ^18^F FDG PET-CT shows promise in assessing residual disease burden following NAC; however, it must be acknowledged that much of the literature comprises small cohort studies, and a negative PET-CT does not confirm the absence of residual disease following NAC. Further large prospective randomised trials are needed to fully validate the findings.

### 4.5. Identifying Recurrence

Up to 50% of MIBC patients develop distant recurrence after radical cystectomy, especially those with advanced or lymph node-positive disease [6]. Studies indicate that ^18^F FDG PET-CT offers high diagnostic accuracy for recurrence, with Alongi et al. reporting a sensitivity of 87%, specificity of 94%, and significant prognostic value for both overall and progression-free survival [42]. Zattoni et al. found that for both MIBC and UTUC, ^18^F FDG PET-CT outperforms CT and MRI, particularly for non-urinary recurrences, with 94% sensitivity and 79% specificity in detecting recurrent urothelial cancer [43]. A meta-analysis by Xue et al. incorporating seven studies with 603 patients reported a pooled sensitivity of 94% and specificity of 91% for ^18^F FDG PET-CT in identifying recurrent or residual disease, often aided by delayed imaging [44].

Summary: Overall, ^18^F FDG PET-CT is highly sensitive and specific for detecting recurrence in urothelial cancer, although its role in routine follow-up post-cystectomy is still under evaluation. Further studies are required to ascertain whether earlier detection of recurrent urothelial cancer confers any improvement in overall survival.

### 4.6. Impact of ^18^F FDG PET-CT on Management

Studies indicate that ^18^F FDG PET-CT detects more malignant lesions than conventional CT or MRI in 20–40% of cases, and it may alter the clinical management plan for up to 68% of patients, with upstaging of disease being more common than downstaging [45,46]. Kibel et al. reported that PET-CT identified metastases in 7 out of 42 patients that were missed in pre-operative evaluations, demonstrating sensitivity and specificity values of 70% and 94%, respectively [35].

A larger retrospective analysis of 711 patients, published in 2022 by Voskuilen et al., found that ^18^F FDG PET-CT potentially influences the treatment of nearly one-fifth (18%) of patients [47]. For half of these patients, it resulted in a shift from potentially curative to palliative treatment. They recommended the routine use of ^18^F FDG PET-CT as part of bladder cancer staging. Al Zubaidi et al. found that ^18^F FDG PET-CT identified metastatic disease missed by conventional CT in approximately 10% of patients, resulting in significant changes in their treatment plans [27]. In this study, patients underwent both standard CT scans of the chest, abdomen, and pelvis, as well as ^18^F FDG PET-CT, with both scans conducted within an 8-week period and prior to NAC. A total of 7 out of 75 patients (9.3%) avoided cystectomy because ^18^F FDG PET-CT detected metastases that CT did not. The study concluded that ^18^F FDG PET-CT might provide additional diagnostic accuracy over CT in patients with muscle-invasive bladder cancer, potentially increasing the sensitivity for detecting nodal or metastatic disease.

Ongoing studies, such as the phase II EFFORT-MIBC trial, continue to investigate the impact of ^18^F FDG PET-CT on patient management. These studies aim to provide additional data that could support the integration of ^18^F FDG PET-CT into current clinical guidelines for risk stratification and treatment adaption in muscle-invasive bladder cancer [48].

Summary: ^18^F FDG PET-CT use can impact on patient management and lead to significant changes in management.

## 5. Alternative Metabolic Tracers

In an effort to overcome the limitation of urinary excretion of FDG, other metabolic tracers have been investigated in PET imaging for urothelial cancer, including ^11^C choline, ^11^C methionine, and ^11^C acetate.

Evidence for these choline tracers, however, seems slightly less promising. A systematic review by Kim et al. in 2018 looked at the diagnostic accuracy of ^11^C choline and ^11^C acetate PET-CT for detecting metastatic lymph nodes in patients with bladder cancer demonstrating low sensitivity and moderate specificity [49]. A comparative study by Golan et al. found^11^C choline PET-CT had no advantage compared to ^18^F FDG PET-CT in detecting metastatic bladder cancer, and concluded that ^18^F FDG PET-CT tended to be more accurate [50]. Similar results were found by Maurer et al. when they compared ^11^C choline PET-CT with contrast enhanced CT in the primary N-staging of 44 bladder cancer patients who subsequently underwent radical cystectomy and lymphadenectomy [51].

^11^C PET tracers, unlike ^18^F labelled agents (such as ^18^F FDG and ^18^F fluoride), appear to have limited clinical value due to their short half-life and the requirement for an on-site cyclotron. Although there is some suggestion that ^11^C methionine could improve tumour detection due to delayed urinary excretion and increased uptake in urothelial lesions without accumulating in inflammatory tissue, its widespread use remains restricted [52].

## 6. Future Directions

### 6.1. FAPI PET

An area of exciting ongoing research in PET imaging is the use of fibroblast activation protein inhibitors (FAPIs), which have demonstrated good results in a range of tumours and appear promising in urothelial cancer. ^68^Ga-labelled FAPI PET specifically targets FAP, which is highly expressed by cancer-associated fibroblasts in the tumour stroma. This tracer rapidly accumulates in lesions while maintaining low background activity, resulting in a high tumour-to-background ratio and improved lesion detection. Unlike many targets, FAP is minimally expressed in healthy tissue, making it a promising candidate for both diagnostic and therapeutic applications [53,54].

Although strong evidence for the use of this novel technique in urothelial cancer is still lacking, several small studies highlight the advantages of ^68^Ga FAPI PET-CT over traditional imaging. Unterrainer et al. found that in a small patient group retrospectively assessed, FAPI PET-CT identified lesions in 26.7% of patients that were missed by conventional CT. Furthermore, in this study, ^68^Ga FAPI PET-CT was able to accurately exclude malignancy for patients previously labelled as having suspicious pelvic lymph nodes and pulmonary nodules on conventional imaging [55,56]. Novruzov et al. demonstrated that ^68^Ga FAPI PET-CT had a higher tumour uptake and mean SUVmax than ^18^F FDG PET-CT, with an additional nine lesions detected exclusively by FAPI PET [57]. Similarly, a small study by Koshkin et al. noted that FAPI PET-CT detected small nodal metastases in patients with both localised and metastatic urothelial cancer, that would not otherwise meet Response Evaluation Criteria In Solid Tumours (RECIST) criteria for malignancy, impacting clinical decisions [58,59]. These findings indicate that FAPI PET-CT could enhance staging accuracy and guide treatment.

### 6.2. Carbonic Anhydrase IX (CAIX)

CAIX is expressed in 70% to 90% of bladder cancers, but not in normal urothelial tissue [60]. It is a promising tumour marker for urothelial carcinoma and is hoped to be a target for the development of new treatments [61]. Several biodistribution studies have already evaluated ^89^Zirconium (^89^Zr)-labelled girentuximab, an anti-CAIX monoclonal antibody (mAb) in renal cell carcinoma [62,63,64]. A similar approach is being tested in urothelial cancer in the phase 1 ZIPUP trial, which is currently examining the feasibility, safety, and utility of PET-CT using ^89^Zr girentuximab (^89^Zr TLX250) in patients either undergoing pre-operative staging of urothelial carcinoma or bladder cancer for curative intent, or with known metastatic urothelial carcinoma. ^89^Zr TLX250 may have utility in the accurate staging of bladder and urothelial carcinomas, with less renal excretion compared to FDG [65].

Hofman et al. recently published a first-in-human study looking at the safety, imaging, and dosimetry of a CAIX-targeting peptide, ^68^Ga DPI-4452, in patients with clear cell renal cell carcinoma [66]. This showed excellent results in terms of tumour uptake in these patients with high tumour to background ratios. While this study did not look at patients with urothelial cancer, there may be potential utility for it in this regard given the high expression of CAIX in urothelial malignancy.

### 6.3. Nectin-4

Nectin-4, a tumour-associated antigen, has been shown to be selectively overexpressed in urothelial cancer, present in approximately 83% of bladder cancers and up to two-thirds of upper tract urothelial cancers, and associated with a poor prognosis [67,68]. Enfortumab Vedotin, an antibody–drug conjugate (ADC) that targets Nectin-4, was approved by the FDA in 2019 for treating locally advanced or metastatic urothelial cancer [69]. One of the major challenges is correctly identifying patients who would benefit from EV therapy, as the expression of Nectin-4 at the level of the primary tumour and metastases determines the amount of the drug that can reach tumour cells [70].

A previous pre-clinical and biodistribution study by Campbell et al. showed promising results with a ^89^Zr-labelled mAb against Nectin-4 as an ImmunoPET probe [71]. More recently, Duan et al. synthesised the molecular probe ^68^Ga N188 and conducted a first-in-human study looking at its use in targeting Nectin-4 for PET-CT imaging in advanced urothelial cancer [72]. They reported a clear correlation between PET SUV value and Nectin-4 expression, suggesting that ^68^Ga N188 PET can be used as a companion diagnostic tool for optimising treatments that target Nectin-4. This probe was compatible with the physical half-lives of commonly used radionuclides.

An iodine-labelled ADC PET probe ^124^I-EV has also been developed with high specificity and binding affinity of Nectin-4. This probe completely simulated the circulation of ADC drugs, including tumour uptake and retention, and so may be of value in the clinical setting [73].

### 6.4. uPAR

PET imaging targeting urokinase plasminogen activator receptor (uPAR) has shown promising results in various cancers. uPAR is expressed in approximately 89% of urothelial cancers and may also be associated with a worse prognosis [74,75]. A number of small studies have looked at the safety, pharmokinetics, and biodistribution of uPAR on PET imaging, including in patients with bladder cancer. ^64^Cu DOTA-AE105 showed a favourable tumour-to-background ratio in both primary tumour lesions and metastatic lymph nodes in bladder cancer patients, providing strong evidence for the potential for uPAR PET-CT in bladder cancer patients [76].

MNPR-1 is an mAb that selectively targets uPAR, and is an exciting prospect in the field of urothelial cancer theranostics. It is currently recruiting for a phase 1 clinical trial in Australia. It is hypothesised that the radioactive-labelled mAb MNPR-101-Zr (MNPR-101 conjugated to ^89^Zr) will be effective at locating tumours with PET imaging [77].

### 6.5. TROP-2

Trophoblast cell surface antigen 2 (Trop-2) has been shown to have increased expression in urothelial cancer and positively correlates with disease severity [78]. Sacituzumab govitecan (IMMU-132), a Trop-2-directed ADC, received approval for treatment of locally advanced or metastatic urothelial cancer in patients previously treated with platinum-based chemotherapy and an ICI-based following findings from cohort 1 of the Trophy-U-01 study [79]. A phase 2 study has recently confirmed initial results and shown the objective response rate remained high in this cohort of patients [80].

### 6.6. EGFR

Epidermal growth factor receptor (EGFR) is overexpressed in bladder cancer with a positive correlation between levels of EGFR, progression, and prognosis [81]. A preclinical study has looked at immunoPET to detect EGFR expression in bladder cancer, utilising a radioimmunoconjugate ^89^Zr-DFO-Panitumumab to specifically target and visualise EGFR in orthotopic bladder tumours. This study showed that the immunoPET signal correlated with tumour volume in the models. It provides a proof of concept, but further studies are necessary to fully validate its potential [82].

### 6.7. HER2

Human epidermal growth factor receptor 2 (HER2) overexpression, well-documented in breast and gastric cancers as a driver of aggressive tumour biology and poor prognosis, is also significant in urothelial cancer, which has the third highest rate of HER2 overexpression among cancers [83]. Approximately 10–20% of urothelial tumours exhibit HER2 amplification or overexpression, underscoring its potential as both a therapeutic target and an imaging biomarker [83]. Several HER2-targeted therapies have been investigated in advanced or metastatic urothelial carcinoma, including mAbs (trastuzumab, pertuzumab), tyrosine kinase inhibitors (lapatinib, afatinib, neratinib), and ADCs (trastuzumab emtansine, trastuzumab duocarmazine). HER2-targeted PET imaging, utilising radiolabelled agents such as ^89^Zr-labelled trastuzumab, presents an opportunity to refine patient selection for HER2-directed therapies. This approach could enhance staging accuracy, identify HER2-positive lesions for personalised treatment, and evaluate therapeutic responses. While these developments are promising, further research and validation are required before they can be adopted into routine clinical practice. Notably, two ongoing trials in this area, highlighted in Table 2, aim to provide additional evidence to support their clinical integration.

### 6.8. PET-MRI

Combining the functional imaging capabilities of PET with the anatomical detail of MRI has the potential to enhance local staging in urothelial cancer, and may detect metastatic bladder cancer lesions not seen on CT or in patients who cannot receive intravenous iodine contrast. Preliminary studies by Li et al. and Civelek et al. suggest that FDG PET-MRI performs well in bladder cancer staging, NAC response prediction, and can detect metastatic lesions, which CT misses [15,84,85]. Overall, there have only been a small number of studies researching PET-MRI in urothelial cancer, and they were limited by small cohorts. Further research is warranted in this area to determine its utility in clinical practice.

## 7. Theranostics and Precision Medicine

Cancer theranostics is a novel approach that combines diagnostic imaging and radionuclide therapy [86]. A major challenge in advancing precision oncology lies in identifying and validating tumour-specific targets that drive cancer progression, with the goal of maximising therapeutic effectiveness while minimising off-target toxicity.

Immuno-positron emission tomography (ImmunoPET) is an advanced imaging technique that combines the specificity of mAbs with the high sensitivity of PET scans [87]. This enables precise imaging of dysregulated pathways in cancerous tissues. Recent advancements in tumour molecular characterisation have identified new molecular targets and biomarkers on tumour cells, opening new avenues for targeted cancer therapies. Monoclonal antibodies labelled with radionuclides form so-called radio-immunoconjugates, which can deliver high-dose radiation directly to the cancer cell. Radioimmunoconjugates have been shown to be useful for both the detection of tumours and for therapy, and can be combined with conventional therapies to enhance the therapeutic efficacy of mAbs [88].

Both beta- (β) and alpha (α)-emitting radioisotopes can be linked to antibodies, allowing delivery of radiation to cells. Traditionally, beta emitters have been used for therapeutic applications due to their negative charge, low linear energy transfer, and long travel distance. Clinically used β emitters include ^177^Lutetium (^177^Lu), ^131^Iodine (^131^I), and ^90^Yttrium (^90^Y). Long-range electrons can travel through several cancer cells, thus increasing the average dose to a tumour; however, the low linear energy transfer means that beta radiation cannot deliver a lethal dose to a targeted single cancer cell. The long range of electrons also results in a larger dose to surrounding healthy tissue.

Alpha particles, on the other hand, are positively charged and larger than β particles. As a result, they have a much higher linear energy transfer than β particles and travel a much shorter distance. The short range of very energetic α emissions in tissue means that a large fraction of that radiative energy travels into the targeted cancer cells [58]. The high linear energy transfer also results in a greater probability of generating DNA double-strand breaks on interaction with cell nuclei. This makes them particularly effective for treating small clusters of cancer cells or micro-metastases [89].

ImmunoPET provides a non-invasive method to assess the in vivo expression and distribution of these targets and shows promise for clinical applications, including diagnosis, selection of targeted therapies, evaluating response to therapy, prediction of adverse events, and dose titration for radioimmunotherapy [90]. Table 2 highlights recent trials in PET imaging for novel tracers in urothelial cancer.

## 8. Radiotheranostics in Urothelial Carcinoma

Radiotheranostics has rapidly evolved to address the unique challenges of urothelial cancer, offering opportunities for both precise diagnosis and targeted therapy. Modern approaches are increasingly moving toward smaller molecules, such as bi-peptides, which offer distinct advantages over radiolabelled antibodies [91]. For example, ^68^Ga N188, a bi-peptide targeting Nectin-4, provides rapid clearance from circulation, resulting in lower background signal and improved imaging quality. Its small size allows greater tissue penetration, enhancing tumour targeting even in dense or heterogeneous tumour environments. Furthermore, the rapid pharmacokinetics of these smaller molecules make them well-suited for pairing with short-lived isotopes such as ^68^Ga, streamlining imaging workflows and reducing patient radiation exposure [72].

In the context of urothelial cancer, these smaller molecules not only offer improved imaging precision, but also hold potential for theranostic applications by delivering therapeutic radionuclides to tumours with minimal off-target effects. By combining faster clearance, greater penetration, and reduced immunogenicity, bi-peptides like ^68^Ga N188 represent an emerging, highly effective approach to radiotheranostics in urothelial cancer. These advances highlight the shift toward next-generation tracers designed to overcome limitations associated with antibody-based agents, setting the stage for improved diagnostic and therapeutic outcomes.

## 9. Conclusions

Nuclear medicine and molecular imaging are transforming the management of urothelial cancer by addressing critical gaps in diagnosis, staging, and response evaluation. ^18^F FDG PET-CT, in particular, has shown advantages over conventional imaging in lymph node and distant metastasis staging, providing valuable prognostic insights that can influence treatment planning and monitoring. ^18^F FDG PET-CT has demonstrated superiority in detecting recurrences post-cystectomy, as well as in assessing response to treatments, suggesting that ^18^F FDG PET-CT could play a more central role in both initial and follow-up imaging of urothelial cancer.

The ongoing development of novel PET tracers is equally promising, with agents like ^68^Ga FAPI showing enhanced sensitivity in targeting fibroblast activation protein in tumour-associated fibroblasts. Similarly, tracers targeting emerging biomarkers, such as Nectin-4 and uPAR, hold potential for improved tumour visualisation and for creating theranostic strategies that combine diagnosis and treatment. As these innovative tracers undergo clinical evaluation, they may expand PET’s role beyond detection, enabling precise, targeted treatments that may reshape urothelial cancer care.

While promising, further research is essential to establish PET’s optimal use in routine practice for urothelial cancer and adoption into clinical guidelines. Future research should focus on validating emerging PET tracers, such as ^68^Ga FAPI and Nectin-4-targeting agents, through large, prospective, multicentre trials to ensure their clinical utility in urothelial cancer. Integration with complementary modalities like PET-MRI should be explored to enhance tumour visualisation and staging accuracy. Developing tracers that combine diagnostic and therapeutic capabilities will further strengthen the role of theranostics in personalised cancer care. Overcoming challenges related to tracer production, standardisation, and urinary excretion will be key to optimising imaging protocols. These efforts will be instrumental in translating innovative PET tracers into routine clinical practice, ultimately improving patient outcomes in urothelial cancer management, and potentially improving survival and quality of life for patients with this challenging disease.

## Figures and Tables

**Table 1 cancers-17-00232-t001:** Pharmacokinetic and selectivity criteria for a successful PET tracer for urothelial cancer.

Pharmacokinetic and Selectivity Criteria for a Successful PET Tracer for Urothelial Cancer:
➢ Suitable structure for radiolabelling
➢ Stable labelling: no dissociation from label during imaging process
➢ Well defined molecular target
➢ High specificity for urothelial cancer biomarkers: FAP, PD-L1, Nectin-4, CAIX, uPAR, etc.
➢ High binding affinity for target
➢ Minimal off target binding; concentrates more in urothelial tumours than in surrounding tissues
➢ Rapid urothelial tumour uptake and clearance from blood
➢ Reduced urinary excretion
➢ Optimal half-life
➢ Capability for Theranostic use
➢ Clinically friendly; cost-effective, non-toxic to patients, easy to manufacture and transport

**Table 2 cancers-17-00232-t002:** Trials in PET imaging for novel tracers in urothelial cancer.

Trial Title	Probe/Tracer	Target	Status (October 24)	Location	Identifier
Nectin-4-Specific LMW PET Probe Imaging in Urothelial Carcinoma	^68^Ga-N188	Nectin-4	Recruiting	China	NCT05321316
Clinical Evaluation of ^18^F-LN1 PET-CT for Imaging of Nectin-4 in Urothelial Carcinomas	^18^F-LN1	Nectin-4	Recruiting	China	NCT06120413
Open Label Pilot Study Evaluating Diagnostic Efficacy and Dosimetry of MNPR-101-DFO*-^89^Zr in Patients with Solid Tumours	MNPR-101-DFO*-^89^Zr	uPAR	Recruiting	Australia	NCT06337084
A Single-centre, Open-label, Phase I Study to Evaluate the Diagnostic Performance of ^89^Zirconium-labelled Girentuximab (^89^Zr-TLX250) PET in Urothelial Cancer Patients (ZiPUP Study)	^89^Zr-TLX250	CAIX	Completed	Australia	NCT05046665
Pilot Study of ^68^Gallium PSMA-PET-CT in Patients With Metastatic Urothelial Carcinoma or Melanoma	^68^Ga PSMA	PSMA	Recruiting	USA	NCT05562791
Expression of Prostate Specific Membrane Antigen (PSMA) in Soft Tissue Sarcomas and Urothelial Cell Carcinomas: Implications for Tumour-specific Molecular Imaging and Treatment?	^18^F-JK-PSMA-7	PSMA	Recruiting	Netherlands	NCT05522257
Prospective Study of ^18^F-HER2 PET in Evaluating the Efficacy of Anti-HER2 Therapy for Urothelial Carcinoma.	^18^F-HER2	HER2	Not yet Recruiting	China	NCT05983796
Al^18^F-HER2-BCH PET-CT to Predict Response in Bladder Cancer Patients Treated with HER2 ADC	Al^18^F-HER2-BCH	HER2	Recruiting	China	NCT06548529
ImmunoPET Targeting Trophoblast Cell-surface Antigen 2 (Trop-2) in Solid Tumours and Compared with ^18^F-FDG	^68^Ga-MY6349	Trop-2	Completed	China	NCT06188468

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
