# Peer review of "Nuclear Medicine and Molecular Imaging in Urothelial Cancer: Current Status and Future Directions"

_cancers, 2025, doi:10.3390/cancers17020232_

Round 1

Reviewer 1 Report

Comments and Suggestions for Authors

This manuscript provides a comprehensive overview of the current and emerging roles of PET imaging in urothelial cancer management. It highlights the challenges and advantages of conventional F-18 FDG PET imaging, compared with CT and MRI. The brief summary of novel PET tracers and theranostic agents is timely and relevant, as these advancements could address existing limitations and improve diagnostic precision. Overall, the manuscript is well-structured and engaging. Detailed comments are as follows:

1.     Please spell out all abbreviations the first time they are shown. Also avoid repeating the spelling out.

2.     Please correct the formatting of radionuclides in the main text and tables. Use the superscript formatting for the numbers when using 18F, 89Zr, etc.

3.     Remove the comment from Page 2, line 88.

4.     Cite the Tables in the main text.

5.     It is recommended to convert Table 1 to a paragraph of writing.

6.     Line 163: correct the formatting issue with “Voskuilen et al”.

7.     Line 188: please correct “it is”.

8.     Line 211: missing the “-“ between PETCT.

9.     Section 7 Theranostics and Precision Medicine: this part is a rather general introduction to radiotheranostics. It is recommended that the authors provide some perspectives on the specific applications of radiotheranostics in urothelial cancer.

10.  Her2 was included in Table 2. It is recommended that the authors provide some analyses and perspectives.

11.  Line 425: formatting issue.

12.  In the conclusion section, it is recommended that the authors provide actionable insights or recommendations for future research to make the new tracers successful in imaging and radiotherapy of urothelial cancer.

13.  Since radionuclide therapy is included in the review and is an important direction for the field, “radiopharmaceuticals”, “nuclear medicine and molecular imaging” or others may be more appropriate for the title instead of “PET imaging”. 

Author Response

Thank you for your detailed evaluation of our manuscript, "PET Imaging in Urothelial Cancer: Current Status and Future Directions in Imaging and Theranostics" (Manuscript ID: cancers-3381006). 

We sincerely appreciate the feedback provided and have carefully considered all comments and made the necessary revisions to the manuscript. Please see the attachment for a response to each comment. 

Reviewer 2 Report

Comments and Suggestions for Authors

The review, titled “PET Imaging in Urothelial Cancer: Current Status and Future Directions in Imaging and Theranostics,” provides a detailed overview of PET imaging's role in urothelial cancer, emphasizing [18F]FDG PET regional staging, alternative metabolic tracers, and future directions. While the review addresses relevant aspects, it requires significant revisions.

My specific comments

·The sentence “Urothelial cancer is the second most common urological malignancy in developed countries” is repeated in the abstract, simple summary, and introduction. Please rewrite this information to avoid redundancy.

·Ensure uniformity in the presentation of PET-related terms throughout the manuscript. For instance, [18F]FDG and "F-18 FDG" appear interchangeably; adopt a single consistent style, such as "[18F]FDG." Similarly, "PET-CT" (line 29) and "PET/CT" (line 30) are used inconsistently; choose one format for consistency. Check and correct the line spacing from lines 34 to 38.

·The introduction should include a paragraph discussing the global incidence of bladder cancer and urothelial carcinoma to provide better context and significance of the topic. A suggested reference for this information is https://seer.cancer.gov/statfacts/html/urinb.html.  

·In lines 62–63, references are missing for the statement: "In recent years, 18F-fluorodeoxyglucose (F-18 FDG) PET-CT has gained traction for staging muscle invasive bladder cancer (MIBC), especially in cases where conventional imaging results are inconclusive.

· Ensure all trials listed in Table 2 include proper citations in a "References" tab. The manuscript lacks a clear description or discussion of Table 1 and Table 2. 

·The manuscript mentions the drawback of urinary tracer excretion affecting bladder tumor imaging. Including few published urothelial cancer PET/MR images to demonstrate how this limitation might be mitigated would significantly enhance the discussion.

Comments on the Quality of English Language

The manuscript would benefit from improved uniformity in line spacing and overall formatting to enhance readability.

Author Response

(The authors gave the same response as above.)

Reviewer 3 Report

Comments and Suggestions for Authors

The paper under review is an interesting study on the role of PET in urothelial tumors. The topic is current, as PET imaging is not yet fully incorporated into international guidelines in this context.
Some suggestions:

  • I suggest that the authors refine the structure of the document and, instead of adding a summary at the end of each paragraph, create a final “discussion” section. This section should highlight the key aspects and challenges of FDG PET in different clinical settings (staging, recurrence, etc.) and could include a table for clarity.
  • Regarding FAPI PET, the authors state: “FAP is minimally expressed in healthy tissue, making it a promising candidate for both diagnostic and therapeutic applications.” This is certainly a relevant aspect. I suggest adding some references on the topic (e.g., doi: 10.3390/ijms24043863).
  • In the “theranostics” section, the authors focus their discussion on radiolabeled antibodies. However, more modern approaches tend to use smaller molecules, such as bi-peptides (e.g., Ga68-N188 for Nectin-4, which the authors mention). These molecules offer faster clearance and reduced (or absent) immunogenicity. Please elaborate on this.
  • Lastly, I do not understand why PET/MRI is included in the “precision medicine” section. It is a hybrid imaging modality that contributes to personalized medicine, like PET/CT, although it is less widespread for various reasons. Please explain and elaborate further.

Author Response

(The authors gave the same response as above.)

Round 2

Reviewer 3 Report

Comments and Suggestions for Authors

The authors have properly addressed Reviewer's concerns.